# A Vegetarian Diet Significantly Changes Plasma Kynurenine Concentrations

**DOI:** 10.3390/biom13020391

**Published:** 2023-02-18

**Authors:** Anne-Lise Bjørke-Monsen, Kristin Varsi, Arve Ulvik, Sunniva Todnem Sakkestad, Per Magne Ueland

**Affiliations:** 1Laboratory of Medical Biochemistry, Innlandet Hospital Trust, 2609 Lillehammer, Norway; 2Laboratory of Medical Biochemistry, Førde Central Hospital, 6812 Førde, Norway; 3Department of Medical Biochemistry and Pharmacology, Haukeland University Hospital, 5021 Bergen, Norway; 4Bevital AS, 5021 Bergen, Norway; 5Department of Medicine, Haukeland University Hospital, 5021 Bergen, Norway

**Keywords:** tryptophan, kynurenines, vegetarian, omnivore, vitamin B6, C-reactive protein, liver, alanine transaminase

## Abstract

Tryptophan is an essential amino acid and a precursor of a number of physiologically important metabolites, including serotonin, melatonin, tryptamine, and kynurenines. We assessed tryptophan, kynurenines, and vitamin B2 and B6, as well as biomarkers of liver function and inflammation, in a group of 158 female omnivores and vegetarians aged 18–40 years. The majority of women were omnivores, and 22% were vegetarians. Vegetarians had 25% lower serum ALT, significantly higher pyridoxal concentrations, and significantly lower plasma concentrations of most kynurenines, varying from 8% lower concentrations of median plasma kynurenine to 42% lower concentrations of plasma xanthurenic acid, compared to omnivores. No significant differences were observed in vitamin B2 status or in inflammation markers, C-reactive protein and neopterin between the groups. Vegetarians had lower levels of several plasma kynurenines compared to omnivores. The reason for this is unknown; however, lower ALT concentrations, suggesting a better liver status, and a more favourable vitamin B6 status might be contributing factors.

## 1. Introduction

Tryptophan (Trp) is an essential amino acid and precursor of a number of physiologically important metabolites, including serotonin, melatonin, tryptamine, and kynurenines [1]. Changes in Trp metabolites have been associated with both immune dysfunction and central nervous system disorders [1]. Trp is abundant in dairy products, meat, fish, eggs, bananas, oats, pumpkin and sesame seeds, chocolate, dried dates, soy, tofu, and nuts [2]. Trp concentrations have been reported to be the highest in male fish-eaters and vegetarians, followed by meat-eaters, and the lowest in vegans [3].

Trp is mainly metabolised to kynurenine (Kyn) through the hepatic kynurenine pathway (HKP) and catalysed by tryptophan 2,3-dioxygenase (TDO), which is responsible for 95% of Kyn metabolism (Figure 1). A minor part is controlled by the enzyme indole 2,3-dioxygenase (IDO), located in a variety of tissues and stimulated by proinflammatory cytokines, such as interferon gamma, tumour necrosis factor alpha (TNF-a), and interleukin-6. TDO is regulated by glucocorticoids, the substrate Trp, and the cofactor heme and by feedback inhibition by reduced nicotinamide adenine dinucleotide (phosphate) (NAD(*p*)H). TDO and IDO activity determines the rate of Trp degradation and the availability of Kyn metabolites for various functions [1].

Kyn is metabolised by the enzyme kynurenine aminotransferase (KAT) to kynurenic acid (KA) or by kynurenine 3-monooxygenase (KMO) to 3-hydroxykynurenine (HK). HK can be converted to xanthurenic acid (XA) by KAT or metabolised through 3- hydroxyanthranilic acid (HAA) by kynureninase (KYNU) to either picolinic acid (Pic) or quinolinic acid (QA). QA is further metabolised by quinolinate phosphoribosyl transferase (QPRT) to nicotinamide adenine dinucleotide (NAD). Kyn can also be converted to anthranilic acid (AA) by the enzyme KYNU.

The enzymes KAT and KYNU require vitamin B6 in the form of pyridoxal 5′-phosphate (PLP) as a cofactor, whereas KMO requires vitamin B2 in the form of FAD as a cofactor. Deficiency of either vitamins causes disturbances in kynurenine metabolites [4]. Significant differences in kynurenine metabolites have been reported according to various diets, particularly between vegetarians and omnivores [5]. We investigated markers of liver function and inflammation, B vitamins, involved in the kynurenine pathway in healthy, never-pregnant women aged 18 to 40 years, of which the majority were omnivores and 22% were vegetarians or vegans.

## 2. Materials and Methods

### 2.1. Study Population and Design

This was an observational study designed to investigate metabolites in the kynurenine pathway, including vitamins B6 and B2, as well as markers of liver function and inflammation, according to the diets of never-pregnant women of fertile age and pregnant women. As there was just one vegetarian among the pregnant women, only the never-pregnant women were included in this work. Between June 2012 and May 2015, healthy, never-pregnant women aged 18 to 40 years were recruited among employees and students at Haukeland University Hospital and the University of Bergen, Norway. Additionally, never-pregnant vegetarians were recruited through Facebook and complied with Facebook.com’s terms of service for recruiting participants on their website (www.facebook.com/policies/ads/, accessed on 15 September 2015). Ethical approval of the protocol was granted by the Regional Committee for Medical Research Ethics West (2011/2447). Written informed consent was obtained from all women, and all methods were performed in accordance with the relevant guidelines and regulations.

### 2.2. Clinical Data

The women completed a questionnaire concerning age, body weight, health status, diet, and the use of micronutrient supplements, alcohol, and tobacco. Regular use of supplements was defined as use for more than three days per week, and the definition of a regular tobacco user was based on a plasma cotinine concentration > 85 nmol/L. This cut-off is commonly used to define both regular smokers [6] and the use of smokeless tobacco [7].

### 2.3. Blood Sampling and Analysis

Non-fasting blood samples were obtained via antecubital venipuncture, collected into Vacutainer Tubes with and without EDTA (Becton Dickinson, Franklin Lakes, NJ, USA), and placed in ice water, and plasma and serum were separated within 4 h. The samples were stored at –80 °C until analysis.

The analysis of liver enzymes and function parameters, i.e., alanine transaminase (ALT), *alkaline phosphatase* (ALP), gamma-glutamyl transferase (GGT), total bilirubin, and albumin, as well as C-reactive protein (CRP), was performed in a routine laboratory using a Roche Cobas c702 analyser (Roche, Mannheim, Germany). Liver enzymes and bilirubin were analysed in all vegetarians (n = 34) and in 59 of the 124 omnivores.

Plasma 1-methylhistidine, 3-methylhistidine, and trimethylamine N-oxide (TMAO) are considered to be good biomarkers of meat and fish intake [8] and were included to assess adherence to the reported diet.

The analysis of methylhistidines, TMAO, neopterin, B2 and B6 vitamers, Trp, and kynurenines were performed via liquid chromatography on an Agilent series 1100 HPLC system coupled with electrospray ionisation tandem mass spectrometry (ESI-MS/MS) on an API 4000 triple-quadrupole tandem mass spectrometer from Applied Biosystems/MSD SCIEX [9]. Plasma levels of cotinine were assayed using the LC-MS/MS method [10].

Pic was added to an existing method based on LC-MS/MS [9] using multiple reaction monitoring (MRM) in the positive mode, recoding the ion pairs 124 *m*/*z*-78 *m*/*z* for picolinic acid and 128 *m*/*z*-88 *m*/*z* for picolinic-d4 acid (internal standard). The within- and between-day CVs were 6–7% and 5–8%, respectively (Appendix A).

QA was added to an existing method based on LC-MS/MS [9] using multiple reaction monitoring (MRM) in the positive mode, recoding the ion pairs 168 *m*/*z*-78 *m*/*z* for quinolinic acid and 171 *m*/*z*-81 *m*/*z* for quinolinic-d3 acid (internal standard). The within- and between-day CVs were 7.0–7.5% and 7.2–10.3%, respectively (Appendix A).

### 2.4. Statistical Analysis

Results are presented with the median and interquartile range (IQR) and were compared via the Mann–Whitney U test, whereas the chi-squared test was used for categorical data.

Spearman’s correlation and multiple linear regression were used to explore relationships between parameters. Predictors of tryptophan and kynurenines were evaluated in multiple linear regression models, which included age, BMI, diet, the use of micronutrient supplements, alcohol, tobacco (plasma cotinine), serum CRP, plasma PLP, and FMN as independent variables.

The SPSS statistical program (version 26) was used for the statistical analyses. Two-sided *p*-values < 0.05 were considered statistically significant.

## 3. Results

### 3.1. Demographics

The study included 158 never-pregnant women, of which 124 had an omnivore diet, 34 were vegetarians, and 4 were vegans. The vegetarian diet was adhered to for a median of 36 months (IQR 16, 132) with a range of 2–240 months.

Micronutrient supplements were used on a regular basis by approximately one fourth of the women, and omega 3 fatty acids were the most popular supplements in both groups. More vegetarians than omnivores used tobacco, based on a plasma cotinine level ≥ 85 nmol/L, but no other significant differences in demographic data were observed between the groups (Table 1).

### 3.2. Biomarkers of Inflammation and Liver Function According to Diet

All three biomarkers of animal food intake, 1-methylhistidine, 3-methylhistidine, and TMAO, were significantly lower in vegetarians compared to omnivores (Table 2).

Neither median serum CRP nor plasma neopterine was significantly different in vegetarians compared to omnivores (Table 2).

Vegetarians had significantly lower ALT and GGT concentrations compared to omnivores, and no significant differences according to diet were observed in serum albumin, liver enzymes, and function (Table 2). There were no significant differences in liver enzymes according to the use of alcohol or smoking in either the vegetarian (*p* > 0.07) or the omnivore group (*p* > 0.08), and serum ALT did not correlate with intake of alcohol (rho: 0.06, *p* = 0.59) in the whole group.

### 3.3. Vitamins According to Diet

Plasma concentrations of B2 and B6 vitamers according to diet are given in Table 3. Vitamin B2 analysis included riboflavin and the flavin cofactor, flavin mononucleotide (FMN) [11], and the major circulating B-6 vitamers included the active form pyridoxal-5′-phosphate (PLP), the transport form pyridoxal (PL), and the catabolite 4-pyridoxic acid (PA) [4]. Median plasma PLP did not differ, but median plasma PL (+41%) and PA (+37%) were higher, and the PLP/PL ratio was lower (−33%) in vegetarians compared to omnivores (Table 3). However, in the multiple linear regression models, which additionally included age, BMI, use of alcohol and tobacco, and serum CRP, both a vegetarian diet and the regular use of micronutrient supplements were significantly associated with higher plasma levels of PLP, PL, and PA (Table 4), and no effects on plasma riboflavin and FMN were observed (data not shown). The ratio between PA/(PLP + pyridoxal) ratio (PAr), a suggested marker of altered vitamin B-6 homeostasis during cellular immune activation [4], was slightly higher in vegetarians compared to omnivores (Table 3).

### 3.4. Kynurenines According to Diet

All median plasma kynurenine concentrations, except for AA, were significantly lower in vegetarians compared to omnivores (Table 3), ranging from 42% lower XA to 8% lower kynurenine, AA, and QA concentrations. No differences were seen for tryptophan, HKr (a marker of B6 status), and KA/QA ratio in relation to diet (Table 3).

The most significant predictor of plasma Trp was the number of alcohol units consumed per week (B: 2, 95% CI for B: 0.7, 2.7, *p* = 0.001), followed by serum CRP (B: −1, 95% CI for B: −2, 0, *p* = 0.06) and omnivore vs. vegetarian diet (B: −5, 95% CI for B: −10, 0, *p* = 0.07) in the multiple linear regression model, which additionally included age, BMI, use of micronutrient supplements, and plasma cotinine.

Being a vegetarian and the use of micronutrient supplements were both positive predictors for all B6 vitamers in the multiple linear regression models, which additionally included age, BMI, use of alcohol, tobacco (plasma cotinine), and serum CRP as independent variables.

Being a vegetarian was also the strongest predictor of all kynurenines, except for AA (*p* = 0.06) and QA, in the multiple linear regression models, which additionally included age, BMI, use of micronutrient supplements, alcohol and tobacco (plasma cotinine), and serum CRP as independent variables. Serum CRP was a significant positive predictor for plasma HK and QA and a significant negative predictor for plasma KA and Pic (Table 4).

Including serum ALT in the regression model did not essentially change the results for B6 vitamers or kynurenines.

There was no significant difference in the KA/QA ratio between omnivores and vegetarians (Table 3). However, in the multiple linear regression model, both serum CRP (B: −1, 95% CI for B: −1.4, −0.5, *p* < 0.001) and being a vegetarian (B: −2.4, 95% CI for B: −4.7, −0, *p* = 0.05) were negative predictors for the KA/QA ratio.

### 3.5. Biomarkers of Inflammation, Liver Function, Vitamins, and Kynurenines in Relation to Smoking

In total, 18 women (11%) were smokers, and the majority of these were vegetarians (Table 1). There were no significant differences between vegetarians and omnivores concerning other demographic variables (*p* > 0.6). Median serum ALT was significantly lower (14 (IQR 11, 19) vs. 19 (15, 22), *p* = 0.02), and so was median serum albumin (46 (IQR 44, 47) vs. 47 (46, 49), *p* = 0.01) and median plasma FMN (9.0 (IQR 7.6, 12.3) vs. 12.0 (9.2, 16.2), *p* = 0.05) in smokers vs. non-smokers, respectively.

Smokers had significantly lower median plasma Kyn (1.31 (IQR 1.11, 1.48) vs. 1.46 (1.31, 1.62), *p* = 0.02), KA (32.3 (IQR 26.9, 41.5) vs. 45.1 (35.8, 57.8), *p* = 0.003), XA (12.8 (IQR 8.2, 21.7) vs. 18.3 (12.7, 25.5), *p* = 0.01), and QA (258 (IQR 224, 328) vs. 332 (283, 392), *p* = 0.004) compared to non-smokers, respectively. However, in the multiple regression models, plasma cotinine was not significantly related to B6 vitamers or kynurenines (Table 3).

## 4. Discussion

Female vegetarians aged 18 to 40 years had significantly lower serum ALT, higher B6 vitamers, and lower levels of all kynurenines apart from AA compared to female omnivores in the same age group.

### 4.1. Biomarkers of Diet

We observed significantly lower plasma concentrations of 1-methylhistidine, 3-methylhistidine, and TMAO in vegetarians compared to omnivores. There is a strong linear relationship between increases in methylhistidine excretion and intake of meat and seafood [12]. Concentrations of 1- and 3-methylhistidine can therefore be used to discriminate between vegetarians and omnivores [12]. TMAO is an oxidation product in the liver from trimethylamine, generated by gut microbiota from carnitine, choline, or choline-containing foods. Most studies that have evaluated the relationship between diet and plasma or urinary concentrations of TMAO seem to indicate that plant-based diets are associated with lower TMAO levels, whereas animal- and seafood-based diets appear to have the opposite effect [13].

### 4.2. Biomakers of Liver Status According to Diet

ALT is generally thought to be specific to the liver; it is, however, also found in the kidney and, in much smaller quantities, in heart and skeletal muscle cells. ALT activity in the liver is about 3000 times that of serum activity, and in the case of hepatocellular injury or death, serum ALT increases [14]. Serum ALT activity is therefore regarded as a reliable and sensitive marker of liver disease and an indicator of overall health, particularly in the context of obesity, metabolic syndrome, and the presence of cardiovascular disease, as many patients affected by these conditions also are at risk of having non-alcoholic fatty liver disease [14]. ALT levels are positively correlated with BMI, glucose levels, and alcohol consumption [15] and are negatively correlated with smoking, physical activity, age, and the use of oral contraceptives [16]. Plant-based diets are reported to be associated with more favourable liver function test profiles [17] and lower likelihoods of fatty liver disease. Our data are in accordance with this. The vegetarians had a 25% lower median serum ALT and a 7% lower GT concentration compared to omnivores. Vegetarian diets are rich in antioxidants and are the main sources of anti-inflammatory phytochemicals, which may protect against non-alcoholic fatty liver disease [17].

### 4.3. Biomarkers of Inflammation According to Diet

A vegetarian diet was reported to be associated with reduced serum CRP concentrations compared to an omnivore diet [18]. We found no significant differences in serum CRP, plasma neopterin, or PAr between vegetarians and omnivores.

### 4.4. Vitamin B6 and B2 According to Diet

A vegetarian diet and the use of micronutrient supplements were associated with higher vitamin B6 status, with particularly higher plasma PL and PA, whereas neither diet nor supplements had any effect on vitamin B2 status. An American population-based study reported that the use of vitamin B6 supplements was the strongest predictor of vitamin B6 status, followed by dietary vitamin B6 intake [19]. The geometric mean of plasma PLP concentrations in vegetarians is 58.2 (±standard error: 6.0) compared to 51.0 (±1.1) nmol/L in meat-eaters [19], resembling our data (9% lower plasma PLP in omnivores compared to vegetarians).

In healthy people, plasma PLP appears to be determined primarily by the intake of vitamin B-6, its binding to albumin, and its hydrolysis to PL by alkaline phosphatase [4]. The liver is the organ responsible for the formation of PLP in plasma [20], and the dephosphorylation of PLP to PL is catalysed by tissue non-specific ALP located on the outer membrane of cells from many organs, including erythrocytes [21].

Median plasma PLP was equal between vegetarians and omnivores, but due to higher PL levels, vegetarians had a significant lower PLP/PL ratio compared to omnivores. The plasma PLP/PL ratio was reported to be lower in critically ill patients with systemic inflammation when compared to controls [22]. However, enhanced ALP activity was also reported to have this effect, in which more PLP dephosphorylated to PL, the membrane transfer form, with more ending up in erythrocytes to be rephosphorylated in their active form [22]. We did not, however, find any difference in serum ALP between vegetarians and omnivores.

Vegetarians also had higher PA concentrations, the vitamin B6 catabolite that is excreted into the urine [23]. Higher level of PL in plasma was reported to increase oxidation to 4′-pyridoxic acid in the kidneys, as confirmed by a reported positive correlation between plasma PL and urinary 4′-pyridoxic acid [22].

### 4.5. Tryptophan and Kynurenines According to Diet

Healthier diets, such as the Mediterranean diet, have been associated with lower concentrations of kynurenines [24]. Significant differences in kynurenine metabolites between vegetarians and omnivores were also reported in an Indian study [5]. In our population of young females, vegetarians had markedly lower levels of all kynurenines, apart from AA, compared to omnivores, and being a vegetarian was the strongest negative predictor for plasma kynurenine concentrations in the multiple linear regression models. The reason for this is unknown, but it could be related to better liver function, lower plasma Trp concentrations, or higher levels of B6 vitamers. The lower ALT concentration in vegetarians may imply that the vegetarians had a better liver status compared to omnivores. As more than 95% of dietary Trp is metabolised in the hepatic kynurenine pathway [1], a better liver status might speed up this pathway and therefore contribute to lower levels of all kynurenines.

Biomarkers of inflammation are associated with dietary quality and have also been associated with changed levels of plasma kynurenines [25]. In our population, serum CRP was associated with lower plasma concentrations of HK and KA in the multiple linear regression models, but none of the inflammation markers differed between vegetarians and omnivores. Weekly intake of alcohol was the strongest positive predictor for plasma Trp, which was a surprising finding. Alcohol consumption has increased by approximately 40% in Norway during the last 20 years, particularly among women, who preferably drink wine [26]. Red wine was reported to contain tryptophan-ethylesters [27], which may be hydrolysed by intestinal mucosa, the liver, and kidneys to provide tryptophan [28]. Hence, tryptophan-ethylesters in red wine may provide a pool of tryptophan that is able to cross the gastrointestinal tract. Concentrations of tryptophan-ethylester in wine are low, reported to be around 0.008 mg/dL. As the recommended daily allowance for tryptophan in adults is estimated to be in the range of 250–425 mg/day, the consumption of red wine can hardly cover the daily needs of tryptophan. Alcohol consumption may, however, be a confounder, as cheese, peanuts, and chocolate [29] are good sources of tryptophan, popular foods among women [30] and commonly used together with wine. Alcohol intake is known to reduce liver function even in young people [31]. However, the women in our study reported a modest intake of alcohol, and this was not related to serum ALT.

### 4.6. Effect of Smoking on Biomarkers and Kynurenines

Cigarette smoke exposure was reported to impact the immune system [32]; however, we observed no difference in inflammatory markers according to smoking. Smokers had lower levels of ALT and albumin, as reported by others [16,33].

The published data show that smokers have lower levels of trp and most kynurenines compared to non-smokers [34]. In our female population, several kynurenines were reduced in smokers compared to non-smokers, particularly plasma KA (reduced by 28%) and plasma QA (reduced by 22%). Smoking was, however, more common among the vegetarians, and in the multiple regression models, which included diet, plasma cotinine was not significantly related to any kynurenines.

### 4.7. Kynurenines and Mood Disorders

Changes in kynurenine metabolites have been implicated in the pathogenesis of mood disorders [35] and schizophrenia [36]. There are both immunactive and neuroactive metabolites in the kynurenic pathway, including the *N*-methyl-d-aspartate (NMDA) receptor antagonist KA and agonist QA [37]. The excessive activation of NMDA receptors may result in excitotoxic neuronal damage; this process has been implicated in the pathomechanism of different neurodegenerative disorders, such as Alzheimer’s disease [38].

A recent meta-analysis reported that kynurenic acid and KA/QA ratio are decreased in mood disorders, such as major depressive disorder, bipolar disorder, and schizoaffective disorders [39]. Plasma KA was significantly lower in vegetarians compared to omnivores, but we saw no difference in the KA/QA ratio between the two groups; however, being a vegetarian and serum CRP were both negative predictors for the KA/QA ratio in the multiple linear regression models.

### 4.8. Strengths and Limitations

The number of vegetarians in this study was small; it was, however, large enough to show distinctive metabolic differences in both vitamin B6 and kynurenine metabolism between vegetarians and omnivores. We did not have extensive details regarding diets, which limited our ability to study how major components of diet may interact with Kyn metabolism.

The intake of alcohol was, quite unexpectedly, a positive predictor of plasma Trp. We did not have information about what type of alcohol the women used, so we were unable to say if the intake of red wine was of any importance. However, this observation was most likely a confounder, which needs further study.

## 5. Conclusions

Female vegetarians aged 18 to 40 years had significantly lower levels of all kynurenines, apart from AA, compared to omnivores in the same age group. Vegetarians had better liver function, as determined by significantly lower serum AL, and higher levels of plasma PL, the membrane transfer form of vitamin B6, compared to omnivores. Both factors might affect kynurenine metabolism, but this needs further study.

## Figures and Tables

**Figure 1 biomolecules-13-00391-f001:**
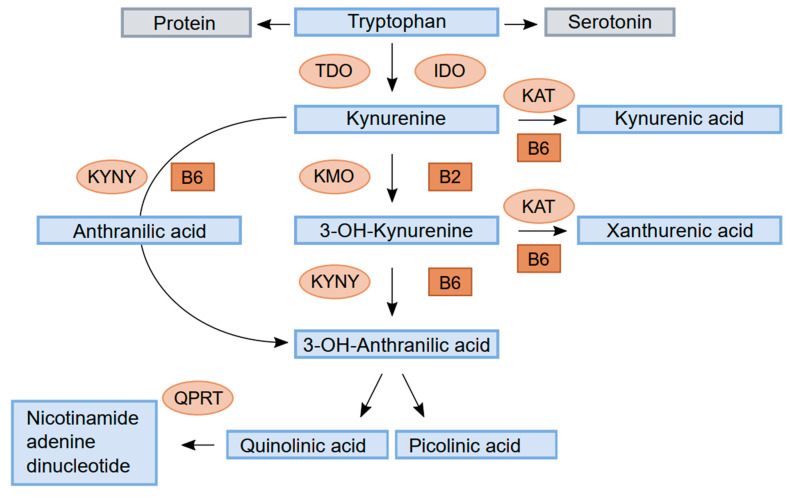
Schematic drawing of tryptophan metabolism through the kynurenine pathway. Enzymes: tryptophan 2,3-dioxygenase (TDO), indole 2,3-dioxygenase (IDO), kynurenine aminotransferase (KAT), kynurenine 3-monooxygenase (KMO), kynureninase (KYNU), and quinolinate phosphoribosyl transferase (QPRT). Cofactors: vitamin B6 (pyridoxal 5′-phosphate, PLP) and vitamin B2.

**Table 1 biomolecules-13-00391-t001:** Baseline characteristics of healthy women according to diet (n = 158).

	Omnivore Diet n = 124	Vegetarian Diet n = 34	*p* Value
Age, years, median (IQR)	24 (22–28)	24 (22–28)	0.65 ^1^
BMI, kg/m ^2^, median (IQR)	22 (21–24)	22 (21–23)	0.88 ^1^
Users of oral contraceptives, n (%)	71 (55%)	17 (50%)	0.60 ^2^
Regular Users of Supplements, n (%)			
Omega 3 fatty acids	59 (48%)	11 (33%)	0.14 ^2^
Multivitamins/minerals	27 (22%)	8 (24%)	0.85 ^2^
Alcohol, number of units/week, median (IQR)	2.0 (0.5–4.0)	2.0 (0.8–3.0)	0.99 ^1^
Smokers ^3^, n (%)	9 (7%)	9 (27%)	0.002 ^2^

^1^ Comparison with Mann–Whitney U test. ^2^ Comparison with Pearson’s chi-squared test. ^3^ Smokers defined as plasma cotinine ≥ 85 nmol/L.

**Table 2 biomolecules-13-00391-t002:** Biomarkers of diet, inflammation, and liver function according to diet in healthy never-pregnant women aged 18–40 years (n = 158).

Median, (IQR)	Omnivore Diet N = 124	Vegetarian Diet N = 34	*p* Value ^1^
Biomarkers of Diet (Plasma)			
1-Methylhistidine, µmol/L	3.62 (3.02–3.98)	2.71 (2.46–3.07)	<0.001
3-Methylhistidine, µmol/L	2.76 (1.16–7.15)	0.33 (0.29–0.40)	<0.001
Trimethylamine-N-oxide, µmol/L	2.72 (2.02–4.05)	1.86 (1.42–2.59)	<0.001
Biomarkers of Inflammation			
Serum-C-Reactive Protein, mg/L	1.0 (1.0–2.0)	0.8 (0.5–2.3)	0.07
Plasma neopterin, nmol/L	11.3 (8.9–14.7)	12.9 (10.0–14.4)	0.31
Biomarkers of Liver Status (Serum)			
Alanine transaminase, U/L ^2^	20 (16–23)	15 (12–19)	<0.001
Alkaline phosphatase, U/L ^2^	59 (51–66)	57 (46–70)	0.54
Gamma-glutamyl transferase, U/L ^2^	14 (12–20)	13 (10–15)	0.04
Total bilirubin, µmol/L ^2^	8 (5–11)	7 (5–9)	0.31
Albumin, g/L	47 (45–48)	46 (45–48)	0.47

^1^ Comparison with Mann–Whitney U test. ^2^ Reduced number of omnivores; N = 59

**Table 3 biomolecules-13-00391-t003:** Plasma concentrations of vitamins and kynurenine metabolites according to diet in healthy never-pregnant women aged 18–40 years (n = 158).

Median, (IQR)	Omnivore Diet N = 124	Vegetarian Diet N = 34	*p* Value ^1^
Vitamins (Plasma)			
Pyridoxal 5-phosphate, nmol/L	63.4 (49.1–98.5)	68.9 (55.6–127.3)	0.20
Pyridoxal, nmol/L	11.6 (9.1–15.6)	16.2 (10.9–36.8)	0.001
4-Pyridoxic acid, nmol/L	21.8 (16.3–28.9)	27.4 (18.3–49.2)	0.02
Pyridoxal 5-phosphate/Pyridoxal ratio	5.8 (4.7–6.8)	4.0 (3.7–5.1)	<0.001
PAr ^2^	28.7 (21.9–36.1)	31.0 (23.7–39.3)	0.14
Riboflavin, nmol/L	8.2 (5.8–13.0)	10.1 (6.3–18.1)	0.20
Flavin mononucleotide, nmol/L	12.0 (9.2–16.4)	11.0 (7.5–13.5)	0.05
Tryptophan/Kynurenine Metabolites (Plasma)			
Tryptophan, µmol/L	70.9 (61.6–81.9)	66.6 (60.2–75.1)	0.13
Kynurenine, µmol/L	1.47 (1.32–1.65)	1.33 (1.18–1.48)	0.001
3-Hydroxykynurenine, nmol/L	42.8 (35.4–52.2)	33.3 (30.1–40.3)	<0.001
Kynurenic acid, nmol/L	46.4 (36.2–60.4)	35.4 (38.7–41.4)	<0.001
Anthranilic acid, nmol/L	12.8 (10.5–15.8)	11.7 (9.8–13.8)	0.09
3-Hydroxyanthranilic acid, nmol/L	47.6 (39.2–60.8)	31.8 (26.8–40.8)	<0.001
Xanthurenic acid, nmol/L	19.1 (14.1–27.2)	10.9 (9.8–17.1)	<0.001
Picolinic acid, nmol/L	58.2 (46.1–73.6)	47.6 (37.3–58.9)	0.003
Quinolinic acid, nmol/L	338 (283–394)	298 (245–330)	0.008
Nicotinamide, nmol/L	194 (144–242)	154 (115–182)	<0.001
HKr ^3^	33 (28–38)	34 (29–43)	0.23
Kynurenic acid/Quinolinic acid ^4^	13 (10–18)	11 (10–16)	0.10

^1^ Comparison with Mann–Whitney test. ^2^ PAr: 4-Pyridoxic acid/(Pyridoxal 5-phosphate + Pyridoxal). The ratio was multiplied by 100. ^3^ HKr: 3-Hydroxykynurenine/(Kynurenic acid + Anthranilic acid + 3-Hydroxyanthranilic acid + Xanthurenic acid). The ratio was multiplied by 100. ^4^ The ratio was multiplied by 100.

**Table 4 biomolecules-13-00391-t004:** Determinants of plasma B6 vitamers and kynurenine concentrations with multiple linear regression ^1^.

Independent Variables	Pyridoxal 5-Phosphate, nmol/L			Pyridoxal, nmol/L			4-Pyridoxic Acid, nmol/L		
	B	95%CI for B	*p* Value	B	95%CI for B	*p* Value	B	95%CI for B	*p* Value
Omnivore vs. vegetarian diet	22.2	1.9, 42.4	0.03	11.9	7.1, 16.6	<0.001	16.5	8.7, 24.3	<0.001
Use of micronutrient supplements ^2^	32.8	11.8, 53.9	0.002	8.2	3.2, 13.1	0.001	17.1	9.0, 25.3	<0.001
Serum C-Reactive Protein ^3^	−3.6	−7.8, −0.5	0.09	−0.3	−1.3, 13.0	0.51	−0	−2, 1	0.61
Plasma cotinine ^3^	−0	−0, 0	0.71	0	−0, 0	0.82	0	−0, 0	0.65
Independent Variables	Kynurenine, µmol/L			3-Hydroxykynurenine, nmol/L			Kynurenic Acid, nmol/L		
	B	95%CI for B	*p* value	B	95%CI for B	*p* value	B	95%CI for B	*p* value
Omnivore vs. vegetarian diet	−0.2	−0.3, −0.1	0.003	−10	−15, −4	<0.001	−12	−19, −6	<0.001
Use of micronutrient supplements ^2^	0.1	−0.1, 0.2	0.34	0.6	−5, 6	0.84	4	−3, 11	0.25
Serum C-Reactive Protein ^3^	−0	−0, 0.2	0.75	1	0, 2	0.04	−3	−4, −1	<0.001
Plasma cotinine ^3^	0	0, 0	0.08	−0	−0, 0	0.09	−0	−0, 0	0.30
Independent Variables	Anthranilic Acid, nmol/L			3-Hydroxyanthranilic Acid, nmol/L			Xanthurenic Acid, nmol/L		
	B	95%CI for B	*p* value	B	95%CI for B	*p* value	B	95%CI for B	*p* value
Omnivore vs. vegetarian diet	−2	−4, 0.1	0.06	−17	−24, −9	<0.001	−8	−12, −4	<0.001
Use of micronutrient supplements ^2^	0.5	−1.5, 2.5	0.62	6	−2, 14	0.14	2	−2, 6	0.36
Serum C-Reactive Protein ^3^	−0.1	−0.5, 0.3	0.52	−0	−2, 1	0.86	0	−1, 1	0.68
Plasma cotinine ^3^	−0	−0, 0	0.62	−0	−0, 0	0.35	−0	−0, 0	0.09
Independent Variables	Picolinic Acid, nmol/L			Quinolinic Acid, nmol/L			Nicotinamide, nmol/L		
	B	95%CI for B	*p* value	B	95%CI for B	*p* value	B	95%CI for B	*p* value
Omnivore vs. vegetarian diet	−15	−25, −6	0.002	−38	−104, 27	0.25	−65	−118, −11	0.02
Use of micronutrient supplements ^2^	4	−6, 14	0.38	44	−24, 113	0.20	5	−50, 60	0.86
Serum C-Reactive Protein ^3^	−2	−4, −0.5	0.01	16	3, 30	0.02	−3	−14, 8	0.62
Plasma cotinine ^3^	−0	−0, 0	0.65	−0.1	−0.1, 0	0.27	−0	−0.1, 0	0.30

^1^ The regression models contain age, body mass index, and use of alcohol (units per week) as independent variables, in addition to the parameters listed in the table. ^2^ Use of micronutrient supplements: non-regular use vs. regular use (≥3 days per week). ^3^ Serum CRP and plasma cotinine as continuous variables.

## Data Availability

Data can be made available from the authors upon reasonable request.

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
