# Peer review of "A Vegetarian Diet Significantly Changes Plasma Kynurenine Concentrations"

_biomolecules, 2023, doi:10.3390/biom13020391_

Round 1

Reviewer 1 Report

In this manuscript, the authors address that female vegetarians aged 18 to 40 had significantly lower kynurenine levels compared to omnivores in the same age group.  

The relationship of plasma kynurenine level between vegetarians and omnivores described in this paper is more straightforward. However, some questions remain concerning a very basic point. 

Minor points

1.     The reason why only women were considered should be clearly stated. In other words, it would be better to clearly state the reason why the study was not conducted with men. It would also be useful to investigate gender differences with respect to diet effects on plasma kynurenine concentration. 

2.     In Table 1, the authors report that vegetarians are significantly more likely to be smokers than omnivores. Although smoking has been reported to affect oxidative stress and inflammation (Neurosci Biobehav Rev. 2013), the authors do not seem to have included enough information on the effects of smoking in this cohort. It would be worth noting that this study investigated the kynurenine level between smokers and non-smokers.

 I believe it would be worthy of publication if the authors added a note regarding the simple question above.

Author Response

Thank you so much for nice and valuable comments!

The reason why only women were considered should be clearly stated. In other words, it would be better to clearly state the reason why the study was not conducted with men. It would also be useful to investigate gender differences with respect to diet effects on plasma kynurenine concentration.

Answer: 

We have added more information about study design in the Methods in order to explain why only women were included:

"This was an observational study designed to investigate metabolites in the kynurenine pathway, including vitamins B6 and B2, as well as markers of liver function and inflammation, according to diet in never-pregnant women of fertile age and pregnant women. As there was just one vegetarian among the pregnant women, only the never-pregnant women were included in this work. "

In Table 1, the authors report that vegetarians are significantly more likely to be smokers than omnivores. Although smoking has been reported to affect oxidative stress and inflammation (Neurosci Biobehav Rev. 2013), the authors do not seem to have included enough information on the effects of smoking in this cohort. It would be worth noting that this study investigated the kynurenine level between smokers and non-smokers.

Answer:

Thank you!

We have included a section on the effect of smoking on biomarkers of inflammation, liver function, vitamers and kynurenines. 

Reviewer 2 Report

Article is well written and enjoyable. I didn't find any grammar error.

I have one quesiton and one concern.

The quesiton is. Line 101. WHy liver enzymes and bilirubin were analysed only in 59 out of 124 omnivores?

My concern is about kynurenine' quantitation. Reference method is citation number 9. In the paper I don't find any data on validation and quantitation of picolinic acid and quinolinic while their concentrations are reported in the paper. Can authors clarify on this point?

Author Response

Thank you for valuable and nice comments!

Concerning your questions:

Line 101. WHy liver enzymes and bilirubin were analysed only in 59 out of 124 omnivores?

Answer: In order to keep study cost low, we did not include liver enzymes and bilirubin in the beginning of our study. When we received more funding these analysis were included. 

My concern is about kynurenine' quantitation. Reference method is citation number 9. In the paper I don't find any data on validation and quantitation of picolinic acid and quinolinic while their concentrations are reported in the paper. Can authors clarify on this point?

We have added information in the Methods on validation and quantitation of picolinic acid and quinolinic and included 2 supplemental notes on each analyte.

Reviewer 3 Report

The article is very interesting and well written, but, in my opinion, should include extensive details regarding the type of diet, in respect with the major components of the vegetarian/vegan/omnivore diets adopted by the participants in the study, as the correlation between the nutritional characteristics in terms of carbohydrates and lipids it is also of great importance, along with the protein source. Another limitation of the study is the geographical restrained pattern of the subjects, not being able to compare diverse dietary typologies, and being able to draw conclusions only on a limited area of interest.

The parallel should be made considering the obtained results with other studies performed in different populations, on different types of diets and epigenetic conditions.  

The conclusions should be clear and boldly expressed, in order to emphasize the relevance of the study.

Author Response

Thank you very much for nice comments and valuable suggestions to our paper!

We do not have extensive details regarding the type of diet, so we are not able to study correlation between the nutritional characteristics and kynurenine metabolism. This has been added to the Strength and limitation section.

 Norway has a small and uniform population, so the geographical restrained pattern of the subjects are a limitation to the study.

As published studies on diet and kynurenine metabolism are very scarce, we were not able to compare our results with other studies performed in different populations, on different types of diets and epigenetic conditions.  T